# 5-Bromo-3,4-dihydroxybenzaldehyde Promotes Hair Growth through Activation of Wnt/β-Catenin and Autophagy Pathways and Inhibition of TGF-β Pathways in Dermal Papilla Cells

**DOI:** 10.3390/molecules27072176

**Published:** 2022-03-28

**Authors:** Jung-Il Kang, Youn Kyung Choi, Sang-Chul Han, Hyunwoo Nam, Gilwoo Lee, Ji-Hoon Kang, Young Sang Koh, Jin Won Hyun, Eun-Sook Yoo, Hee-Kyoung Kang

**Affiliations:** 1Jeju Research Center for Natural Medicine, Jeju National University, Jeju 63243, Korea; jikang0024@jejunu.ac.kr (J.-I.K.); choiyk@jejunu.ac.kr (Y.K.C.); yskoh7@jejunu.ac.kr (Y.S.K.); jinwonh@jejunu.ac.kr (J.W.H.); eunsyoo@jejunu.ac.kr (E.-S.Y.); 2Department of Medicine, School of Medicine, Jeju National University, Jeju 63243, Korea; hanschh@jejunu.ac.kr (S.-C.H.); nam97425@naver.com (H.N.); kome4@naver.com (G.L.); jhkang@jejunu.ac.kr (J.-H.K.)

**Keywords:** hair growth, 5-bromo-3,4-dihydroxybenzaldehyde, dermal papilla cells, proliferation, Wnt/β-catenin, autophagy, TGF-β

## Abstract

Various studies addressing the increasing problem of hair loss, using natural products with few side effects, have been conducted. 5-bromo-3,4-dihydroxybenzaldehyde (BDB) exhibited anti-inflammatory effects in mouse models of atopic dermatitis and inhibited UVB-induced oxidative stress in keratinocytes. Here, we investigated its stimulating effect and the underlying mechanism of action on hair growth using rat vibrissa follicles and dermal papilla cells (DPCs), required for the regulation of hair cycle and length. BDB increased the length of hair fibers in rat vibrissa follicles and the proliferation of DPCs, along with causing changes in the levels of cell cycle-related proteins. We investigated whether BDB could trigger anagen-activating signaling pathways, such as the Wnt/β-catenin pathway and autophagy in DPCs. BDB induces activation of the Wnt/β-catenin pathway through the phosphorylation of GSG3β and β-catenin. BDB increased the levels of autophagic vacuoles and autophagy regulatory proteins Atg7, Atg5, Atg16L, and LC3B. We also investigated whether BDB inhibits the TGF-β pathway, which promotes transition to the catagen phase. BDB inhibited the phosphorylation of Smad2 induced by TGF-β1. Thus, BDB can promote hair growth by modulating anagen signaling by activating Wnt/β-catenin and autophagy pathways and inhibiting the TGF-β pathway in DPCs.

## 1. Introduction

Hair loss is defined by symptoms of thinning and decreasing hair on the body and head due to chemotherapy, immune abnormalities, and an imbalance of hormones and nutrients [1,2,3,4]. Although the effect of hair loss on human health is negligible, fear of first impressions and lack of confidence in patients with hair loss increases their desire for treatment. In 2014, more than one million patients with hair loss worldwide underwent surgical or nonsurgical treatment [5]. As of 2015, the global hair loss market was valued at over $7.3 billion [5]. Therefore, researchers have been trying to find a cure for hair loss. 2,4-Diamino-6-piperidinopyrimidine 3-oxide (minoxidil) and 17β-(*N*-*tert*-butylcarbamoyl)-4-aza-5α-androst-1-en-3-one (finasteride) are Food and Drug Administration (FDA)-approved drugs that have been used as therapeutic agents [6,7]. Nevertheless, minoxidil and finasteride have limited use because their effects on hair loss are temporary, and they exhibit side effects, such as infertility [8,9]. Various natural products, including caffeine, procyanidin B2, and vitamin D, are potential therapeutic candidates for hair loss [10,11,12]. However, additional research is needed to identify safer and more scientifically proven materials for hair loss treatment.

Hair follicles are small tissues in the skin where hair growth or loss occurs because of changes in the hair cycle, which consists of the anagen, catagen, and telogen phases [13]. Dermal papilla cells (DPCs) located at the base of hair follicles are present during the hair cycle and produce growth factors that interact with hair follicle cells, including adjacent matrix, dermal sheath, and stem cells [13,14]. These growth factors play different roles depending on the type of cells; in relation to hair growth, transforming growth factor-β (TGF-β) and fibroblast growth factor-5 (FGF-5) act as negative regulators, while vascular endothelial growth factor (VEGF) and FGF-7 act as positive regulators [15]. Additionally, TGF-β, FGF, and VEGF are closely associated with autophagy [16,17,18]. Autophagy is an evolutionarily conserved lysosomal degradation system in eukaryotes that is important for maintaining cellular homeostasis [19,20]. Autophagy inhibition has been reported to be associated with hair follicle regression [21].

In DPCs, the mechanisms of action of minoxidil that have been identified so far include inhibition of progression of the catagen phase by activation of the Wnt/β-catenin pathway and inhibition of apoptosis by activation of Akt and Erk [22,23]. Changes in cell cycle and cell cycle-related proteins are involved in the survival and death of mammalian cells [24,25]. In DPCs, minoxidil exhibits a proliferative effect accompanied by an increase in the levels of cell cycle-related proteins, such as cyclin D1, cdc2, and p34 [26,27].

Interest in compounds found in marine natural products, including green, brown, and red algae, sponges, and marine microorganisms, and their various physiological activities is increasing [28]. In previous studies, 5-bromo-3,4-dihydroxybenzaldehyde (BDB) was isolated from red algae, such as *Polysiphonia morrowii* [29]. BDB exhibits anti-inflammatory effects in LPS-stimulated macrophages and dinitrochlorobenzene (DNCB)-induced atopic dermatitis mouse models and inhibits ultraviolet B (UVB)-induced oxidative stress in keratinocytes [30,31]. In addition, BDB inhibits the production of inflammatory cytokines in bone marrow-derived mast cells [32]. However, the effects of BDB on hair growth remain unknown. This study was conducted to elucidate the effect of BDB on hair growth and its mechanism of action in DPCs, a key regulator of hair growth.

## 2. Results

### 2.1. BDB Promotes an Increase in the Length of Hair Fibers on Cultured Vibrissa Follicles Ex Vivo

To investigate the hair growth effect of BDB, rat vibrissa follicles were isolated, as previously described [33]. Rat vibrissa follicles were cultured in a medium supplemented with BDB or minoxidil for 21 days. The length of hair fibers on the vibrissa follicles treated with minoxidil (10 μM), the positive control group, increased by 136.8% ± 24.0% compared to that in the control group (100% ± 26.6%). Treatment with 0.01, 0.1, and 1 μM BDB increased the length of hair fibers on the vibrissa follicles by 87.31% ± 17.7%, 98.29% ± 26.3%, and 175.7% ± 22.44% (*p* < 0.05), respectively, compared to that in the control group (Figure 1b,c). In particular, 1 μM BDB significantly increased the length of hair fibers on the vibrissa follicles compared to minoxidil (Figure 1b,c).

### 2.2. BDB Increases the Proliferation of DPC

To examine the efficacy of natural products in promoting hair growth or inhibiting hair loss, DPC proliferation was measured, owing to the importance of DPCs as regulators of hair growth and regeneration [14,15,23]. We investigated whether the hair growth effect of BDB was induced by the stimulation of DPC proliferation. DPCs were cultured with various concentrations of BDB or minoxidil for 24–72 h, and cell proliferation was assessed using an MTT assay. After stimulation with BDB (0.001, 0.01, 0.1, 1, and 10 μM) for 48 h, the proliferation of DPCs increased by 104.8% ± 3.8% (*p* < 0.05), 106.3% ± 6.5%, 104.7% ± 3.0% (*p* < 0.05), 105.4% ± 6.3%, and 96.7% ± 5.3%, respectively, compared to the control group (100% ± 3.5%) (Figure 2a). As shown in Figure 2a, BDB (0.001, 0.01, 0.1, 1, and 10 μM) treatment for 72 h increased the proliferation of DPCs by 110.1% ± 8.2% (*p* < 0.01), 111.4% ± 6.2% (*p* < 0.001), 114.1% ± 9.4% (*p* < 0.001), 107.3% ± 7.7% (*p* < 0.05), and 95.4% ± 8.4%, respectively, compared to the control group (100% ± 5.8%). It was also observed that minoxidil, a positive control, significantly increased the proliferation of DPCs by 106.1% ± 4.2% (*p* < 0.05) at 48 h or 109.6% ± 7.6% (*p* < 0.01) at 72 h compared to the control group (Figure 2a). However, BDB and minoxidil did not affect proliferation of DPCs at 24 h (Appendix A). These results demonstrated that BDB can effectively increase DPC proliferation. Cell proliferation and death are associated with changes in cell cycle-related proteins [27,34]. To investigate the mechanism by which BDB promotes DPC proliferation, we examined the levels of cell cycle-related proteins in DPCs. As shown in Figure 2b,c, after treatment with minoxidil for 24 h, the levels of cell cycle-related proteins, such as cyclin E (*p* < 0.05), cyclin D1, phospho-(thr160)-cyclin-dependent kinase-2 (CDK2) (*p* < 0.01), and CDK2, increased, which is consistent with the results of previous studies [26]. In the BDB-treated group, the levels of cyclin E, cyclin D1, phospho-(thr160)-CDK2, and CDK2 increased, similar to the results seen after minoxidil treatment. BDB treatment at both concentrations of 0.01 and 0.1 μM significantly increased the level of phospho-(thr160)-CDK2 (Figure 2b,c). These results suggest that BDB contributes to the increase in the proliferation of DPCs through changes in the levels of cell cycle-related proteins.

### 2.3. BDB Activates the Wnt/β-Catenin Pathway

The Wnt/β-catenin pathway plays an essential role in hair growth, regeneration, and prolongation of the duration of the anagen phase [22,35,36]. Application of minoxidil to mouse skin prolonged the anagen phase of the hair cycle, which was attributed to activation of the Wnt/β-catenin pathway by minoxidil in DPCs [22]. To investigate whether BDB could activate the Wnt/β-catenin pathway in DPCs, the cells were stimulated with BDB (0.01 and 0.1 μM) for 24 h. As a result, BDB treatment significantly increased the level of phospho(ser9)-glycogen synthase kinase3β (GSK3β) at concentrations of both 0.01 and 0.1 μM (Figure 3a,b). At a concentration of 0.01 μM, BDB significantly increased the level of phospho(ser675)-β-catenin, whereas at a concentration of 0.1 μM, BDB significantly increased the level of phospho(ser552)-β-catenin (Figure 3a,b). As shown in Figure 3a,b, minoxidil significantly increased the levels of phospho(ser552)-β-catenin and phospho(ser675)-β-catenin. In addition, increased levels of phospho(ser552)-β-catenin and phospho(ser675)-β-catenin, and increased nuclear translocation were observed using confocal microscopy after 1 h of BDB or minoxidil treatment (Figure 3c,d). These results suggest that, similar to minoxidil, BDB can activate the Wnt/β-catenin pathway in DPCs, which is followed by the prolongation of anagen and hair growth.

### 2.4. BDB Induces the Autophagy in DPCs

Autophagy plays an essential role in hair growth by regulating the anagen phase maintenance during the hair cycle [21]. Therefore, we investigated whether BDB affected autophagy. When DPCs were stimulated with BDB (0.01 and 0.1 μM) or MXD (10 μM) for 24 h, both concentrations of BDB significantly induced autophagy. Moreover, the number of autophagic vacuoles was increased by MXD treatment, which was used as a positive control (Figure 4a). Autophagy is mediated by autophagy-related (Atg) and microtubule-associated protein 1A/1B-light chain 3 (LC3). Most *Atg* genes are required for efficient autophagosome formation and are involved in LC3 conversion (LC3A to LC3B), a major step in autophagosome formation [37]. Studies have shown that hair growth in Atg7-deficient mice is slower than that in controls, suggesting that autophagosome formation is important for hair growth [38]. To determine whether the BDB-induced autophagy was mediated by Atgs and LC3, DPCs were exposed to different concentrations of BDB (0.01 and 0.1 μM) for 24 h. BDB treatment increased the expression levels of Atg7, Atg5, Atg 16 L, and LC3B at both concentrations (Figure 4b,c). MXD (10 μM) treatment, the positive control, also led to an increase in the expression of all autophagy-related proteins (Figure 4b,c). As shown in Figure 4d,e, the treatment with 0.1 μM BDB for 0–24 h led to an increase in the levels of Atg7, Atg5, Atg 16 L, and LC3B compared to that of the control. These results indicated that BDB stimulates autophagy in DPCs and induces hair growth.

### 2.5. BDB Inhibits TGF-β1-Induced Activation of Smad2 in DPCs

The TGF-β pathway is involved in the progression of the hair cycle from the anagen phase to the catagen phase [39]. Therefore, to investigate whether BDB could inhibit the TGF-β pathway in DPCs, cells were stimulated with different concentrations of BDB (0.01 and 0.1 μM) for 24 h. As shown in Figure 5a,b, BDB treatment did not affect the nuclear translocation of Smad2/3, a mediator of the TGF-β pathway. TGF-β1 secreted from balding DPCs is known to induce apoptosis of epithelial cells in hair follicles [40]. To investigate whether TGF-β1-induced activation of Smad could be inhibited by BDB treatment, DPCs were stimulated with TGF-β1 for 1 h after pre-treatment with BDB. TGF-β1 significantly increased the levels of phospho-Smad2 and phospho-Smad3, and the increase in the levels of phospho-Smad2 induced by TGF-β1 was significantly inhibited by BDB (Figure 5c,d). However, BDB did not affect TGF-β1-induced phospho-Smad3 expression. (Figure 5c,d). These results suggest that BDB can inhibit the action of TGF-β1 in DPCs, which is followed by the inhibition of the transition to the catagen phase, thereby prolonging hair growth.

## 3. Discussion

In this study, we showed that BDB derived from red algae induces hair growth by increasing the length of hair fibers in cultured vibrissa follicles ex vivo. We also observed that BDB increased the proliferation of DPC, a hair growth regulator, by regulating cell cycle-related protein levels and sustaining the anagen phase through the activation of Wnt/β-catenin and autophagy pathways and inhibition of the TGF-β pathway.

The demand for hair growth and hair loss prevention products is increasing, owing to concerns about the quality of life and appearance as the environment changes. Among the various models used to identify natural products that show efficacy for hair growth, the ex vivo hair follicle culture model has several advantages. Effective materials can be found in a relatively short period of time, and it is possible to observe which cells in the hair follicle proliferate or die and whether the expression of specific proteins changes, during, or after the experiment [41]. We investigated the hair growth effect of BDB in cultured rat vibrissa follicles and observed that BDB effectively increased the length of hair fibers on vibrissa follicles after 3 weeks. In previous studies, the concentration that showed an effect in the ex vivo model was generally lower than the concentration that showed a proliferative effect on DPCs [26,42]. However, in this study, BDB induced the proliferation of DPCs more effectively at concentrations below 1 μM, but in ex vivo models, BDB did not increase the length of hair follicles at concentrations below 1 μM. These results suggest that BDB has the potential to act on other follicular cells or be converted to active metabolites during 3 weeks of culture. On the other hand, regarding the proliferation, like minoxidil, the DPCs proliferation effect of BDB was observed at 48 and 72 h, but not at 24 h. These results suggest that BDB can promote hair growth by exerting a proliferation effect on DPCs, similar to minoxidil. Cell proliferation is accompanied by changes in cell cycle and cell cycle-related protein expression levels [34]. Among several cell cycle-related proteins in DPCs, BDB significantly increased the level of phospho-CDK2. Phosphorylation of CDK2 induces phosphorylation of Rb and is involved in progression to the S phase of the cell cycle [34,43]. These results suggested that BDB increased the proliferation of DPCs through cell cycle progression.

Hair growth is positively or negatively regulated by various signaling factors. The mechanisms by which BDB promotes hair growth and proliferation of DPCs in this study are as follows: First, the Wnt/β-catenin pathway plays an important role in hair growth as well as in the regulation of proliferation of DPCs [22,27,36]. It has been reported that the Wnt/β-catenin pathways are involved in hair regeneration after wounding, prolongation of the anagen phase, and inhibition of apoptosis in DPCs [22,35]. In DPCs, minoxidil activates the Wnt/β-catenin pathway to regulate the expression of Wnt/β-catenin target genes such as Axin2 and Lef-1 [22]. In another study, minoxidil induced PKB phosphorylation and inhibited the apoptosis of DPCs [23,44]. BDB induces phosphorylation/stabilization of β-catenin by increasing the level of phospho(ser9)-GSK3β. This, in turn, induces the translocation of β-catenin into the nucleus, suggesting that BDB-mediated DPC proliferation can be induced through the activation of the Wnt/β-catenin pathway. Second, several recent studies have reported the relationship between autophagy and hair growth. According to a previous study, autophagic structures in scalp hair follicles tended to decrease in the early/middle catagen phase than in the anagen phase [21]. In addition, many small molecules (α-ketoglutarate, oligomycin, 5-aminoimidazole-4-carboxamide ribonucleotide, metformin, and rapamycin) that are known to induce autophagy have been reported to promote hair growth [45]. In this study, BDB promoted the expression of Atg7, Atg5, Atg16L, and LC3, suggesting that an increase in autophagy induced by BDB could affect the induction of DPC proliferation. Finally, the TGF-β pathway inhibits hair growth and proliferation of epithelial cells [39,40]. TGF-β1 injection in mice induces early progression to the catagen phase [39]. Androgen induces TGF-β1 mRNA transcription in DPCs and inhibits the growth of keratinocytes co-cultured with androgen-treated DPCs [40]. BDB inhibited TGF-β1-induced increase in the level of phospho-Smad2, suggesting that this may have a protective effect on the inhibition of TGF-β1-induced apoptosis of keratinocytes and hair loss. On the other hand, BDB has been reported to exhibit antioxidant effects [31]. Some antioxidants such as silymarin and resveratrol show hair growth effects [46,47]. However, we did not investigate the direct correlation between the antioxidant effect of BDB and hair growth in this study, so we will conduct a study to elucidate this in the future. Despite the diverse signaling modulating effects of BDB, to clearly determine the activity of BDB on the hair growth, further studies on other hair-cycle regulation pathway and using various follicular cells are needed.

## 4. Materials and Methods

### 4.1. Reagents

BDB was purchased from Matrix Scientific (Colombia, SC, USA). Dulbecco’s modified Eagle’s medium and fetal bovine serum (FBS) were purchased from HyClone (Logan, UT, USA). Dulbecco’s phosphate-buffered saline (PBS) was obtained from Welgene (Daegu, Korea). Earle’s balanced salt solution (EBSS), hydrocortisone, insulin, 3-[4,5-Dimethylthiazol-2-yl]-2,5-diphenyltetrazolium bromide (MTT), dimethyl sulfoxide (DMSO), and minoxidil were purchased from Sigma-Aldrich (St. Louis, MO, USA). The PRO-PREP protein extraction solution was obtained from iNtRON Biotechnology (Seoul, Korea). NE-PER^TM^ nuclear and cytoplasmic extraction reagents were purchased from Thermo Fisher Scientific (Waltham, MA, USA). Lab-Tek^®^ chamber slides were purchased from Nalgene Nunc International (Rochester, NY, USA). Vectastain mounting medium was purchased from Vector Laboratories (Burlingame, CA, USA). Williams medium E, l-glutamine, and penicillin/streptomycin solution (Pen/Strep) were purchased from Gibco Life Technologies (Grand Island, NY, USA). Westar Nova 2.0 ECL solution was obtained from Cyanagen (Bologna, Italy). Polyvinylidene fluoride (PVDF) membranes were purchased from Bio-Rad (Hercules, CA, USA). X-ray film was purchased from Agfa-Gevaert (Mortsel, Belgium). The Cyto-ID autophagy detection kit was purchased from Enzo Life Sciences (Villeurbance, France).

### 4.2. Animals

Three-week-old male Wistar rats were purchased from Orient Bio (Seongnam, Gyeonggi, Korea) and provided with standard laboratory diet and water ad libitum. All animals were cared for using protocols (approval number: 2015-0014) approved by the Institutional Animal Care and Use Committee (IACUC) of Jeju National University (approval date: 6 August 2015).

### 4.3. Isolation and Culture of Rat Vibrissa Follicles

Wistar rats were euthanized using carbon dioxide (CO_2_). The left and right mystacial pads were separated and placed in E/P buffer containing 1% Pen/Strep. Vibrissa follicles were carefully separated from the mystacial pads under a dissecting microscope to avoid scarring. The separated vibrissa follicles were transferred to a 24-well plate containing William’s E medium (supplemented with 2 mM L-glutamine, 10 μg/mL insulin, 50 nM hydrocortisone, and 1% Pen/Strep) and incubated at 37 °C in a 5% CO_2_ incubator. The medium containing BDB or minoxidil was exchanged once every three days, and photos were taken while culturing for 21 days. The length of the vibrissa follicles was measured using an image analyzer (DP controller; Olympus, Japan), and the growth of the vibrissa follicles was measured by comparing the average value of the change in follicle length with the average length of the control group. 

### 4.4. Cell Culture and Proliferation Assay of Dermal Papilla Cells

Immortalized dermal papilla cells (DPCs) isolated from rat whiskers were cultured in DMEM containing 10% FBS and 1% Pen/Strep at 37 °C in a 5% CO_2_ incubator and subcultured every 3 days. The proliferation of DPCs was measured using the MTT assay. The DPCs (2000 cells/well) were suspended in DMEM containing 1% FBS and incubated in a 96-well plate. After 24 h, cells were treated with or without BDB (0.001, 0.01, 0.1, 1, and 10 μM) or minoxidil (10 μM), a positive control, for 72 h. After incubation with the MTT dye (50 μL/well) for 4 h, the supernatant was removed and formazan was dissolved by adding 200 μL/well of DMSO. The absorbance (540 nm) was measured using a Versamax microplate reader (Molecular Devices, Sunnyvale, CA, USA), and the results were expressed as the percentage change compared with the absorbance value of the control group.

### 4.5. Western Blot Analysis

The DPCs were stimulated with or without BDB (0.01 and 0.1 μM) or minoxidil (10 μM) for 24 h. In some cases, the cells were treated with 0.1 μM of BDB for various times (0–24 h). To investigate whether BDB inhibits the action of TGF-β, cells were treated with BDB (0.1 μM) for 2 h, followed by treatment with TGF-β1 (2 ng/mL) for 1 h. Whole cell lysates were lysed using PRO-PREP protein extraction solution and intracellular (nuclear and cytoplasmic) fractions were separated using NE-PER reagents. The proteins were subjected to 8–12% sodium dodecyl sulphate–polyacrylamide gel electrophoresis (SDS-PAGE) and transferred to polyvinylidene fluoride membranes. The membranes were incubated with 5% non-fat dry milk for 1 h and then incubated with primary antibodies (Appendix A) at 4 °C overnight. The membranes were incubated with the corresponding HFP-conjugated secondary antibodies and bands were detected using Westar Nova 2.0 chemiluminescent reagent. Band intensity was quantified using the ImageJ software (http://rsb.info.nih.gov/ij/, accessed on 1 December 2021).

### 4.6. Immunofluorescent Staining

DPCs (5000 cells/mL) were seeded on Lab-Tek II Nalge chamber slides (Nalge Nunc International, Rochester, NY, USA) in DMEM containing 1% FBS. After 24 h incubation, the cells were stimulated with BDB (0.01 and 0.1 μM) or minoxidil (10 μM) for 1 h. For immunofluorescent staining, the cells were fixed with 4% paraformaldehyde for 15 min, followed by permeabilization with 0.1% Triton^TM^ X-100 for 15 min. The cells were incubated with blocking buffer (0.1% Tween 20-PBS containing 1% BSA and 22.52 mg/mL glycine) for 1 h and then incubated with primary antibodies (phospho-ser552-β-catenin and phospho-ser675-β-catenin, or α-tubulin) at 4 °C overnight. The cells were then incubated with Alexa Fluor^®^ 488- or Alexa Fluor^®^ 594-conjugated secondary antibodies for 1 h. The cells were covered in Vectastain mounting medium containing DAPI, and immunofluorescence images were acquired using a FluoView FV1200 confocal microscope (Olympus, Tokyo, Japan).

### 4.7. Cyto-ID Autophagy Detection Assay

DPCs (5.0 × 10^5^ cells/60 mm dish) were seeded in DMEM containing 1% FBS for 24 h, and then treated with different concentrations of BDB (0.01 and 0.1 μM) or MXD (10 μM) for 24 h. Cells were harvested, washed with PBS, and stained with Cyto-ID dye (1 μL/4 mL assay buffer) for 30 min at room temperature in the dark. The fluorescence intensity of the autophagic vacuoles was analyzed using a FACStar flow cytometer (BD Biosciences, San Jose, CA, USA).

### 4.8. Statistical Analysis

Data were expressed as mean ± standard deviation (SD) or standard error (SE) of experiments performed at least thrice. Statistical significance was determined using GraphPad Prism 7 (GraphPad Software, San Diego, CA, USA). A *p* < 0.05 was considered statistically significant.

## 5. Conclusions

In conclusion, we found that BDB exerts hair growth-promoting effects through various mechanisms in DPCs, which are regulators of hair growth. BDB induces the proliferation of DPCs through changes in cell cycle-related proteins. BDB activated the Wnt/β-catenin and autophagy pathways involved in hair growth, while inhibiting the TGF-β pathway involved in hair loss. Thus, these results suggest that BDB could be used as a therapeutic agent to alleviate hair loss.

## Figures and Tables

**Figure 1 molecules-27-02176-f001:**
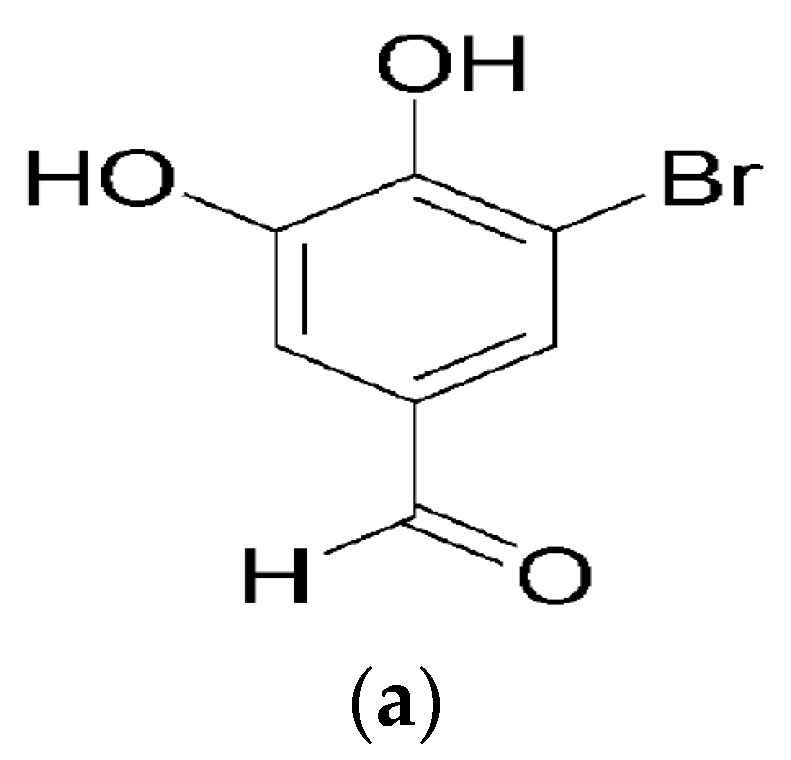
BDB increases the length of hair fibers on the cultured hair follicles ex vivo. Rat vibrissa follicles were stimulated using BDB (0.01, 0.1, and 1 μM) or minoxidil (10 μM) for 21 days: (**a**) Structure of BDB. (**b**) Photograph of rat vibrissa follicles cultured at 0 and 21 days. (**c**) Changes in length of hair fibers on the vibrissa follicles treated with BDB or minoxidil for 21 days. The bar graph shows the growth percentage compared to the average length in the control group on day 21. Each dot indicates independent length of hair fibers on the vibrissa follicle (%). Data are presented as mean ± SD. * *p* < 0.05 compared with the control. The red word (Korean word) in the (**b**) means length.

**Figure 2 molecules-27-02176-f002:**
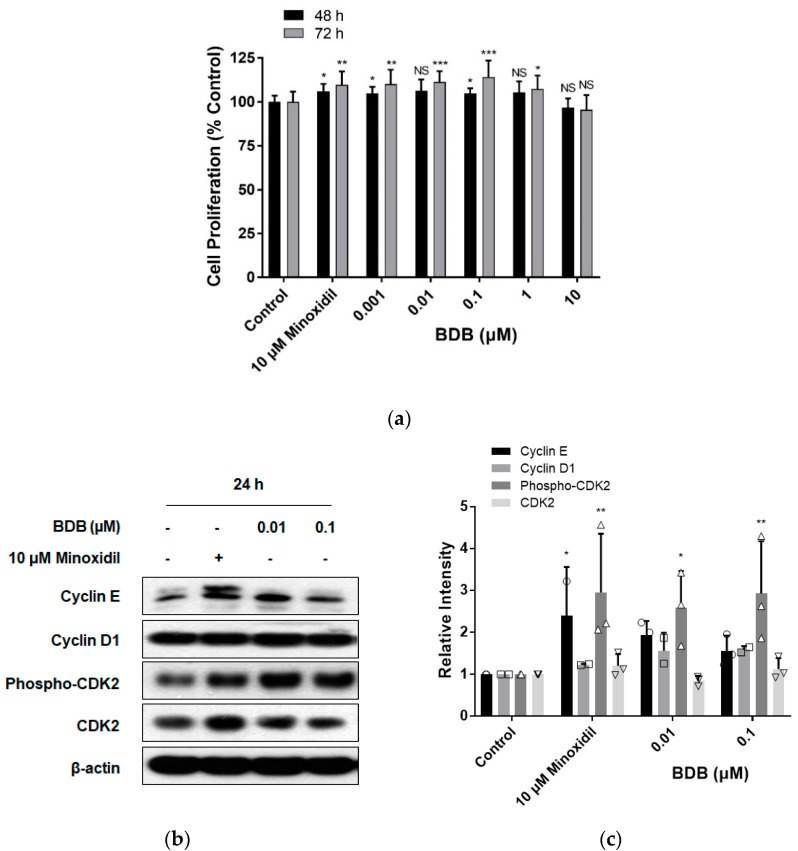
BDB increases the proliferation of DPCs: (**a**) DPC proliferation stimulated using various concentrations of BDB or minoxidil for 48 and 72 h. Data are presented as mean ± SD. * *p* < 0.05, ** *p* < 0.01, *** *p* < 0.001 compared with the control. NS, not significant. (**b**) Changes in the level of cell cycle-related proteins in DPCs treated with BDB or minoxidil for 24 h. (**c**) Quantitative graph of changes in cell cycle-related protein levels after treatment with BDB or minoxidil. Each dot indicates an independent intensity of the protein levels. Data are presented as mean ± SD. * *p* < 0.05, ** *p* < 0.01 compared with the control.

**Figure 3 molecules-27-02176-f003:**
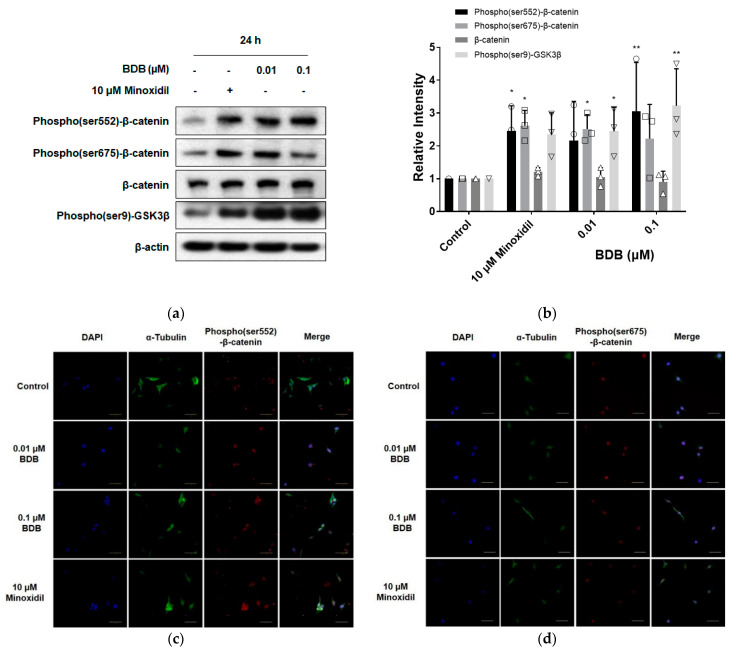
BDB activates the Wnt/β-catenin pathway in DPCs: (**a**) Changes in the level of Wnt/β-catenin proteins in DPCs treated with BDB or minoxidil for 24 h. (**b**) Quantitative graph of Wnt/β-catenin proteins changed by treatment with BDB or minoxidil. Each dot indicates an independent intensity of the protein level. Data are presented as mean ± SD. * *p* < 0.05, ** *p* < 0.01 compared with the control. (**c**,**d**) Intracellular localization of phospho(ser552)-β-catenin and phospho(ser675)-β-catenin observed by confocal microscopy.

**Figure 4 molecules-27-02176-f004:**
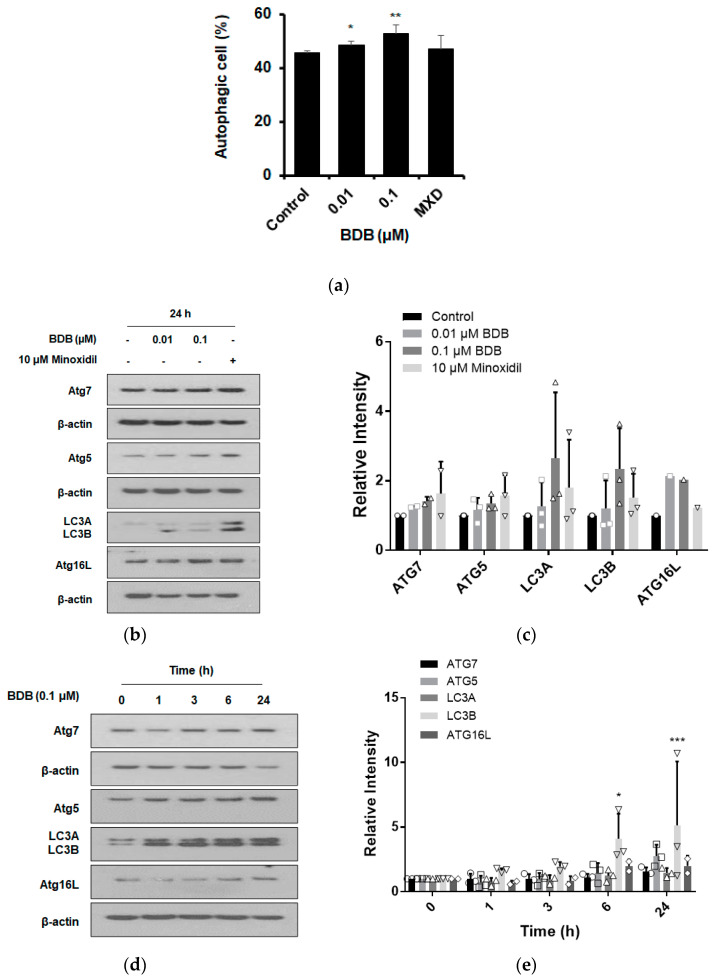
BDB induces autophagosome formation in DPCs: (**a**) Formation of autophagic vacuoles altered by BDB treatment for 24 h. The DPCs were stained with Cyto-ID fluorescence dye for 30 min at room temperature in the dark. Data were analyzed using a FACStar flow cytometer. (**b**) Changes in the expression levels of autophagy-related proteins in DPCs treated with BDB or minoxidil for 24 h. (**c**) Quantitative graph of changes in expression levels of autophagy-related proteins after treatment with BDB or minoxidil. (**d**) Changes in autophagy-related protein levels in DPCs treated with BDB for 0–24 h. (**e**) Quantitative graphs of changes in expression levels of autophagy-related proteins after BDB treatment for 0–24 h. Each dot indicates an independent intensity of the protein level. Data are presented as mean ± SD. * *p* < 0.05, ** *p* < 0.01, *** *p* < 0.001 versus the vehicle (DMSO)-treated control group.

**Figure 5 molecules-27-02176-f005:**
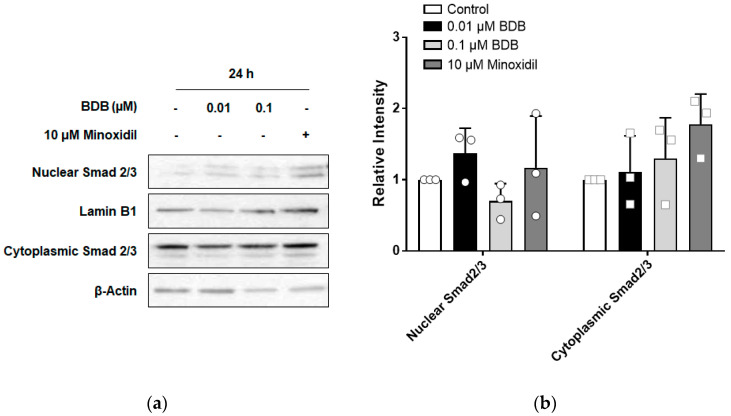
BDB inhibits TGF-β1-induced activation of Smad2 in DPCs: (**a**) Changes in the nuclear/cytoplasmic translocation of Smad2/3 in DPCs treated with BDB or minoxidil for 24 h. (**b**) Quantitative graph of the changes in nuclear/cytoplasmic translocation of Smad2/3 after treatment with BDB or minoxidil. (**c**) Changes in the activation of Smad2 and Samd3 in DPCs treated with BDB and/or TGF-β1. (**d**) Quantitative graph of the changes in activation of Smad2 and Samd3 after treatment with BDB and/or TGF-β1. Each dot indicates an independent intensity of the protein levels. Data are presented as mean ± SD. ** *p* < 0.01, *** *p* < 0.001 compared with the control. ^†††^
*p* < 0.001 compared with the TGF-β1-treated group.

## Data Availability

Not applicable.

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
