# Peer review of "5-Bromo-3,4-dihydroxybenzaldehyde Promotes Hair Growth through Activation of Wnt/β-Catenin and Autophagy Pathways and Inhibition of TGF-β Pathways in Dermal Papilla Cells"

_molecules, 2022, doi:10.3390/molecules27072176_

Round 1

Reviewer 1 Report

The aim of this manuscript entitled BDB promotes hair growth through activation of Wnt/β-catenin and autophagy pathways and inhibition of TGF-β pathways in dermal papilla cells  was to evaluate the stimulating effect of 5-bromo-3,4-dihydroxybenzaldehyde (BDB) and its underlying mechanism of action on hair growth using rat vibrissa follicles and dermal papilla cells (DPCs), required for the regulation of hair cycle and length.

This research is important and can bring valuable information with practical application, and that could be important for the future research. The presented research is well-planned, and the manuscript is well organized. There were used an appropriate and modern experimental design.  Therefore, the work could be of interest, but some points must be considered prior acceptance:

- The title could be corrected. The title could be replaced with 5-bromo-3,4-dihydroxybenzaldehyde promotes hair growth through activation of Wnt/β-catenin and autophagy pathways and inhibition of TGF-β pathways in dermal papilla cells. 

The tested compound, 5-bromo-3,4-dihydroxybenzaldehyde (BDB) is the one isolated from the red alga Polysiphonia morrowii mentioned in the text or it is from another source? It is not very clear in the text.

- The Introduction provides some data on the stage knowledge of this issue. The Introduction data could be supplemented with other information regarding to BDB (another source, pharmacological actions). In the introduction, the design of the research could be presented more clearly, with a presentation of the steps and methods used to answer the research question.

 - There were used an appropriate and modern methodology. The Iconography is appropriate; the figures/tables are well conceived.

 -The discussions interpret the findings in view of the results obtained in this research and the authors take in attention more other literature data related with this subject.

- The conclusions contain the most representative results obtained in this research.

- Thus, the paper overlaps the requirements of this journal.

Reviewer 2 Report

Title: BDB promotes hair growth through activation of Wnt/β-catenin 2 and autophagy pathways and inhibition of TGF-β pathways in 3 dermal papilla cells.

Authors: Kang, et al.

The authors are reporting, in this manuscript, a naturally available small molecule, from red algae, named 5-bromo-3,4-dihydroxybenzaldehyde (BDB), possessing a stimulating effect on hair growth using rat vibrissa follicles and dermal papilla cells (DPCs) which is required for the regulation of hair cycle and length. Mechanistically, the authors claim that BDB promotes hair growth by modulating anagen signaling by activating Wnt/β-catenin and autophagy pathways and inhibiting the TGF-β pathway in DPCs.

Minoxidil (2,4-diamino-6-piperidinopyrimidine 3-oxide)  and finasteride (17β-(N-tert-butylcarbamoyl)-4-aza-5α-androst-1-en-3-one) are FDA-approved drugs that have been used as therapeutic agents [see manuscript references 6,7]. However, they have limited use because their effects on hair loss are moderate, and exhibit side effects, such as infertility [see references 8,9]. Therefore, the motivation of identifying additional natural small molecules safer for hair loss treatment and with less or no side effects is high.

Based on the presented data, the authors concluded that BDB:

  1. exerts hair growth-promoting effects through various mechanisms in DPCs, which are regulators of hair growth.
  2. induces the proliferation of DPCs through changes in cell cycle-related proteins.
  3. activated the Wnt/β-344 catenin and autophagy pathways involved in hair growth, while inhibiting the TGF-β pathway involved in hair loss.

However, these results are not moderately supporting their conclusions. In addition, BDB is highly reactive and unstable which may dump its use as a therapeutic agent to alleviate hair loss.

Additional comments:

DPCs were cultured with various concentrations of BDB or minoxidil for 72 h, and cell proliferation was assessed using an MTT assay. Was DBD assessed at 24 and 48 hrs?

Fig. 2A. It is hard to think that BDB has a significant effect on cell proliferation.

Fig. 2B-C. The error bars are too high. It is hard to confirm that there is a significant effect of BDB  in each data.

Figure 3. c-d. (c-d) confocal microscopy data is not clear.

Overall, the quality of the figure needs to be improved.

Chemically, 5-bromo-3,4-dihydroxybenzaldehyde (DBD) is a fragment small molecule that contains a highly reactive moiety (aldehyde). Therefore, it is highly possible that BDB is metabolized quickly which lowers its use for potential therapeutics. What is the half-life of DBD?

Additionally, with the 3.4-dihydroxy groups attached to the benzene ring, DBD may act as an antioxidant compound that promotes hair growth or inhibit hair loss. Was DBD assessed for its antioxidant properties?

Some acronyms are not defined (e. g., UVB)

Line 96: Figure 2A instead of Figure 1A.

Table 1 may be in the supplementary info.

Round 2

Reviewer 2 Report

The authors kindly updated the manuscript based on the suggested comments. However, no additional experimental data was performed during the revision stage to further support the conclusion of this work.
